# FEW-ROUND LEARNING FOR FEDERATED LEARNING

## ABSTRACT

Federated learning (FL) presents an appealing opportunity for individuals who are willing to make their private data available for building a communal model without revealing their data contents to anyone else. Of central issues that may limit a widespread adoption of FL is the significant communication resources required in the exchange of updated model parameters between the server and individual clients over many communication rounds. In this work, we focus on preparing an initial model that can limit the number of model exchange rounds in FL to some small fixed number $R$. We assume that the tasks of the clients participating in FL are not known in the preparing stage. Following the spirit of meta-learning for few-shot learning, we take a meta-learning strategy to prepare the initial model so that once this *meta-training* phase is over, only $R$ rounds of FL would produce a model that will satisfy the needs of all participating clients. Compared to the meta-training approaches to optimize personalized local models at distributed devices, our method better handles the potential lack of data variability at individual nodes. Extensive experimental results indicate that meta-training geared to few-round learning provides large performance improvements compared to various baselines.

## 1 INTRODUCTION

Major machine learning applications including computer vision and natural language processing are currently supported by central data centers equipped with massive computing resources and ample training data. At the same time, growing amounts of valuable data are also being collected at distributed edge nodes such as mobile phones, wearable client devices and smart vehicles/drones. Directly sending these local data to the central server for model training raises significant privacy concerns. To address this issue, an emerging trend known as federated learning (McMahan et al., 2017; Konecny et al., 2016; Bonawitz et al., 2019; Li et al., 2019; Zhao et al., 2018; Sattler et al., 2019; Reisizadeh et al., 2019), where server uploading of local data is not necessary, has been actively researched.

Unfortunately, federated learning (FL) generally requires numerous communication rounds between the server and the distributed nodes (or clients) for model exchange, to achieve a desired level of prediction performance. This makes the deployment of FL a significant challenge in bandwidth-limited or time-sensitive applications. Especially in real-time applications (e.g., connected vehicles or drones), where the model should quickly adapt to dynamically evolving environments, the requirement on many communication rounds becomes a major bottleneck. Moreover, the considerable amounts of time and computational resources required for training place a high burden on individual clients wishing to participate in FL. Excessive communication rounds in FL are a major concern especially in light of the increased communication burden for guaranteeing full privacy via secure aggregation (Bonawitz et al., 2017).

To combat this limitation, we focus on preparing an initial model that can quickly obtain a high-accuracy global model within only a few communication rounds between the server and the clients. Following the spirit of meta-learning for few-shot learning, we meta-train the model via episodic training to mimic and tee up for *few-round FL*. Meta-training enables reliable prediction even when the data sample at hand does not share the same characteristics with the dataset the given model was trained with. In contrast to existing meta-training attempts to initiate a model for further personalized optimizations at local devices, our approach takes advantage of FL's ability to exploit varying data distributions across clients. A high-level description of our idea is depicted in Fig. 1. Given a small target value $R$, our goal is to create an initial model that can quickly adapt, within $R$ rounds of FL, to a set of clients with tasks not seen during meta-training. As long as the tasks are different

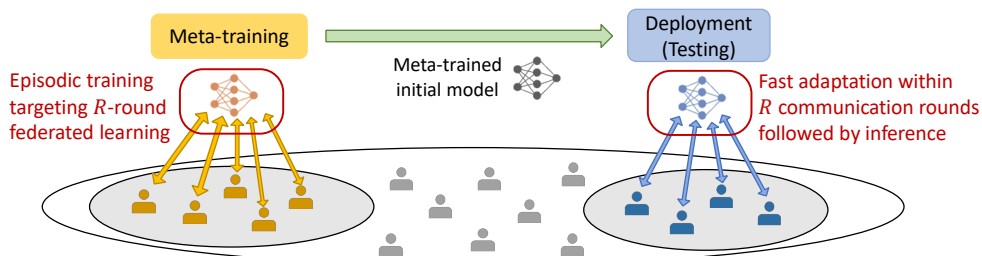

Figure 1: Overall procedure of the proposed few-round learning algorithm for federated learning. By using the meta-trained initial model, a set of clients with new tasks can quickly obtain a high-accuracy global model within only a few rounds of FL. Meta-training is based on episodic training that mimics actual inference preceded by an $R$-round FL procedure. A global prototype-assisted learning strategy at both meta-training and deployment phases further improves model accuracy.

between meta-training and deployment, it is immaterial whether a node that participated in the meta-training also partakes in few-round federated learning (followed by inference). In the context of image classification, different tasks mean classification involving different sets of image categories or classes (e.g., different categories of lung diseases to be diagnosed using chest X-ray images). The prospect of meta-training also raises an intriguing possibility that meta-training can actually be done using proxy data at the server simply mimicking the federated optimization process, although we will not be concerned with this approach in the present work.

Extensive experimental results show that our few-round learning algorithm outperforms various baselines in both IID (independent, identically distributed) and non-IID data distribution setups. In an IID setup, for example, our algorithm achieves a 75.32% the accuracy on *tiered*ImageNet within only $R = 3$ rounds of FL, which surpasses fine-tuned federated averaging (FedAvg) by 14.63% and fine-tuned one-shot federated learning (Guha et al., 2019) by 12.88%.

## 2 RELATED WORKS

**Few-shot learning.** Few-shot learning is an instantiation of meta-learning. In the context of image classification, few-shot learning typically involves episodic training where each episode of training data is arranged into a few training (support) sample images and validation (query) samples to mimic inference that uses only a few examples (Vinyals et al., 2016). Through a repetitive exposure to a series of varying episodes with different sets of image classes, the model learns to handle new tasks (classification against unseen classes) each time. Two widely-known few-shot learning methods with different philosophical twists, which are also conceptually relevant to the present work, are model-agnostic meta-learning (MAML) of (Finn et al., 2017) and Prototypical Networks of (Snell et al., 2017). MAML attempts to generate an initial model from which different models targeting different tasks can be obtained quickly via just a few gradient updates. The idea is that the initial model is learned via meta-training to develop an internal representation that is close in some sense to a variety of unseen tasks. Prototypical Networks, on the other hand, do not rely on such fine-tuning using few examples but rather learn embedding space such that model outputs cluster around class prototypes, the class-specific centroids of the embedder outputs. With episodic training, simple Prototypical Networks appear to be effective in learning inductive bias for successful generalization to new tasks. Our work takes from both concepts: we utilize prototype representation and we also adopt fine-tuning during $R$-round federated learning in adapting to new tasks.

**Federated meta-learning.** Recent research activity has focused on improving model personalization via federated meta-learning (Lin et al., 2020; Chen et al., 2018; Fallah et al., 2020; Jiang et al., 2019). The common goal of these works is to generate an initial model based on which each new client can find its own optimized model via a few local gradient steps and using only its own data. In these works, meta-learning employed during federated learning intends to enable each client to handle previously unseen tasks, in the spirit of MAML of (Finn et al., 2017). User-specific next-word prediction at individual smartphones, for example, could be an application of approaches along this direction. Compared to this line of work, we focus on creating an initial model that leads to a high-accuracy global model, rather than personalized models, within only a few rounds of *federated learning*. In this way, we seek to take advantage of a higher variety of data as well as the larger data volume that would be made available through collaborative learning of many distributed nodes. A clear example is the diagnosis of a broader class of diseases that would be possible through collec-

tive training across numerous examples contributed by a larger group of individuals. Personalized optimization would be especially at disadvantage in non-IID setting, where each client necessarily lacks a sufficient variety of training data.

**One-shot federated learning.** Another line of work recently focused on one-shot federated learning, where the goal is to train a global model with just one communication round between the server and the clients. The authors of (Guha et al., 2019) proposed an ensemble method to choose reliable client-specific models from given clients. To effectively capture global information, they proposed various criteria by which participating clients are chosen. In the work of (Shin et al., 2020), local clients send XOR-encoded MNIST image data to the server, and the server decodes it to train the global model. While the server would need certain data in advance to decode the received results, XOR operation can serve as data augmentation while preserving privacy. In the fusion learning of (Kasturi et al., 2020), each local client uploads both the model parameters and the distribution parameters to the server. The server generates artificial data samples from the distribution parameters to train a global model. When the data gets complex, however, it is not clear whether conversion into a simple distribution would be reliable. Compared to the existing works on one-shot federated learning that employ some randomly initialized model, the key difference of our method is the use of *meta-learning* to obtain an initial model which can adapt to the unseen tasks of individual clients within $R$ rounds of FL. It is shown in Section 4 that the suggested approach yields remarkable performance gains compared to one-shot federated learning methods.

## 3 PROPOSED FEW-ROUND LEARNING ALGORITHM

### 3.1 PROBLEM SETUP

**Federated learning.** Federated learning allows each distributed node $k$ with a dataset $D_k$ to participate in iterative learning of a global model $\theta$ without having to reveal its data to anyone else including the central server. As a given round $r$ starts, each of $K$ participating nodes downloads a global model $\theta_r$ from the server and updates it using its own local data $D_k$. The updated local models $\theta_{r+1}(k)$ get all uploaded to the server, which then aggregates them to get a new model $\theta_{r+1} = \sum_{k=1}^{K} \rho_k \theta_{r+1}(k)$ according to the relative dataset sizes $\rho_k = \frac{|D_k|}{\sum_{k=1}^{K} |D_k|}$. As the next rounds starts, this new global model gets downloaded to all participating nodes and the same process is repeated. Federated learning generally requires a significant number of such global rounds to achieve the desired accuracy, with each round taking up substantial computing and communication resources.

**Problem formulation.** To enable fast federated learning, we would like to prepare an initial model $\phi$ in which a group of clients can quickly obtain a high-performance global model within $R$ communication rounds. In applying meta-learning, we wish to satisfy clients who hope to predict data that were unseen during meta-training. During $R$ rounds of FL at deployment time, each client is also expected to make available some small number of labeled examples associated with the data she hopes to predict. To create a training environment matching the actual $R$-rounds of FL followed by inference at deployment, in each episode, our meta-training updates the model over $R$ federated rounds using the support set ($R$ global rounds) and then makes a final adjustment (meta-update) using the query set. This process is repeated as the model is exposed to a series of episodes.

### 3.2 META-TRAINING

Our meta-training procedure is given in Algorithm 1. Let $\phi_0$ be the initial model and $\phi_T$ the model generated after all $T$ episodic stages. To construct each episode $t$, the server selects a new set of $K$ clients participating for training[1]. The model $\phi_t$ has been updated through the end of episode stage $t-1$. The server lets $\theta_0 = \phi_t$ and sends this initial model to the selected $K$ clients. Through $R$ rounds of local updates and global aggregations guided by the support sets, the model $\theta_0$ evolves to $\theta_R$. Before moving to the next episode, a loss based on the updated model $\theta_R$ and the local query set is minimized with respect to $\phi_t$ to get each meta-updated local model $\phi_{t+1}(k)$. In the final server aggregation of the episode stage, the global meta-updated model $\phi_{t+1}$ is obtained from the local models, which then becomes the initial model for the next episode. Here, $\phi$ is a parameter that is updated by meta-update process with the query sets, and $\theta$ is a parameter that is updated during $R$ rounds of federated learning with the support sets.

---

[1]If the server has enough data, there is no need to utilize the distributed clients for meta-training; the server can simply mimic the federated learning setup and procedure.

Specifically, given $K$ clients in each episode, we first split $D_k$ (the local dataset of client $k$) into support set $S_k$ and query set $Q_k$ such that they are disjoint. Starting from the initial model $\theta_0$, each client $k$ utilizes its support set $S_k$ to update the local model during $R$ global rounds. The local update in Line 16 involves a loss function that depends on class prototypes, as to be clarified shortly. In our method, the server aggregates not only the local models (Line 18) but also the local prototypes (Line 20). After $R$ rounds of local updates and global aggregations, the server obtains the global model $\theta_R$ and sends it to the $K$ clients while still in the current episode stage. Based on the downloaded $\theta_R$ and prototypes, each client $k$ utilizes its query set $Q_k$ to compute the query loss $L_{\text{local}}^{Q_k}$ and meta-updates $\phi_t$ to obtain $\phi_{t+1}(k)$ in Line 24. Note that the local loss $L_{\text{local}}^{Q_k}$ obviously depends on $\theta_R$, which in turn depends on $\theta_0 = \phi_t$; in principle $L_{\text{local}}^{Q_k}$ is minimized with respect to $\phi_t$ in this step.

At last, the server finds the final output $\phi_{t+1}$ of the episode by aggregating the meta-updated local models $\phi_{t+1}(k)$ for $k \in \{1, 2, ..., K\}$. In the meta-learning point of view, the initial model $\phi$ first adapts to the local support sets for $R$ global rounds in the first step, and then is meta-updated based on the losses computed from the adapted model $\theta_R$ and the local query sets, in the second step. This two-step training procedure mimics the deployment phase where the initial model is updated for $R$ communication rounds before making predictions. In the following, we provide some missing details on the description of loss functions and prototype processing.

---

**Algorithm 1** Meta-Training procedure for Proposed Few-Round Learning

---

**Input:** Initialized model $\phi_0$ **Output:** Model $\phi_T$ after $T$ training episodes

1: **for** each training episode $t = 0, 1, ..., T - 1$ **do**
2:     The server constructs an episode with $K$ clients
3:     Each client $k \in \{1, 2, ...K\}$ splits $D_k$ into support set $S_k$ and query set $Q_k$
4:     $\theta_0 \leftarrow \phi_t$
5:     **for** each communication round $r = 0, 1, ..., R - 1$ **do**
6:         **for** each client $k$ **in parallel do**
7:             **if** $r = 0$ **then**
8:                 Download $\theta_r$ from the server
9:             **else**
10:                 Download $\theta_r$ and $P_{r-1}$ from the server
11:             **end if**
12:             **for** each class $c \in C_k$ **do**
13:                 $P_r^c(k) = \frac{1}{|S_k(c)|} \sum_{x \in S_k(c)} f_{\theta_r}(x)$             ▷ Local prototype calculation
14:                                                         with support set $S_k$
15:             **end for**
16:             $\theta_{r+1}(k) \leftarrow \theta_r - \alpha \nabla_{\theta_r} F^{S_k}(\theta_r)$          ▷ Local update of $\theta$ with $S_k$
17:         **end for**
18:         $\theta_{r+1} = \sum_{k=1}^K \lambda_k \theta_{r+1}(k)$          ▷ Server aggregation of local models $\theta$;
19:                                                  $\lambda_k$ is relative support set size
20:         $P_r = \left\{ \sum_{k=1}^K \lambda_k P_r^c(k) | c = 1, 2, ..., N_c \right\}$    ▷ Server aggregation of local prototypes
21:     **end for**
22:     **for** each client $k$ **in parallel do**
23:         Download $\theta_R$ and $P_{R-1}$ from the server
24:         $\phi_{t+1}(k) \leftarrow \phi_t - \beta \nabla_{\phi_t} L_{\text{local}}^{Q_k}(\theta_R, P_{R-1})$.    ▷ Local meta-update of $\phi$ with query set $Q_k$
25:     **end for**
26:     $\phi_{t+1} = \sum_{k=1}^K \rho_k \phi_{t+1}(k)$   ▷ Server aggregation of meta-updated models; $\rho_k$ is relative data size
27: **end for**

---

### 3.2.1 $R$ ROUNDS OF LOCAL UPDATES AND AGGREGATIONS

In defining the loss function, we utilize the class prototypes and associated distance metric of (Snell et al., 2017). For each communication round $r = 0, 1, ..., R - 1$, we not only aggregate the global model $\theta_{r+1}$ but also the set of global prototypes $P_r = \{P_r^c | c = 1, 2, ..., N_c\}$ for all classes $c \in \{1, 2, ...N_c\}$, where $N_c$ is the number of classes in the current episode over all clients.

**Model and global prototype download.** In the beginning of round $r \geq 1$, the server has the global model $\theta_r$ and the set of global prototypes $P_{r-1} = \{P_{r-1}^c | c = 1, 2, ..., N_c\}$ which are the outputs of

the previous round $r - 1$. Each client $k \in \{1, 2, ..., K\}$ first downloads $\theta_r$ and $P_{r-1}$ from the server. Since there is no global prototype in the first round, the clients only download the model $\theta_0$ from the server when $r = 0$.

**Local prototype calculation.** Based on the downloaded model $\theta_r$, each client $k$ computes the local prototype $P_r^c(k)$ for each class $c \in C_k$ using its support set $S_k$ as

$$P_r^c(k) = \frac{1}{|S_k(c)|} \sum_{x \in S_k(c)} f_{\theta_r}(x), \tag{1}$$

where $C_k$ is a set that contains all classes in client $k$ and $S_k(c)$ is a set of all data samples in $S_k$ labeled with class $c$. For a given input data sample $x$, $f_\theta(x)$ represents the output of function $f$ with model parameters $\theta$. This local prototype $P_r^c(k)$ can be viewed as a representative of class $c$ calculated based on the local data (support set) of client $k$.

**Loss calculation from local prototype.** Let $P_r(k)$ be the set that contains all types of prototypes in client $k$, defined as $P_r(k) = \{P_r^c(k) | c \in C_k\}$. Now using $S_k$, $\theta_r$ and $P_r(k)$, each client $k$ computes the local loss $L_{\text{local}}^{S_k}(\theta_r, P_r(k))$ according to

$$L_{\text{local}}^{S_k}(\theta, P_r(k)) = \frac{1}{\sum_{c \in C_k} |S_k(c)|} \sum_{c \in C_k} \sum_{x \in S_k(c)} \{d(f_\theta(x), P_r^c(k)) + \log \sum_{c' \neq c} \exp(-d(f_\theta(x), P_r^{c'}(k)))\}, \tag{2}$$

which is based on the squared euclidean distance $d(\cdot)$ between $P_r^c(k)$ and $f_\theta(x)$ for $x \in S_k(c)$. We would like to minimize this local loss during local update at each client.

**Auxiliary loss from global prototype.** If we only consider the loss function of (2), each client would have biased models after local updates, especially when the data distributions across different clients are non-IID. This generally leads to a performance degradation of the global model. Hence, we propose a global prototype-assisted learning (GPAL) strategy where a global prototype helps to train the local models in the right direction; the set of global prototypes $P_{r-1}$ can encourage local models to maintain stability during optimization because the global prototype contains all the information in local data from the entire set of clients. We adopt the dense classification loss of (Lifchitz et al., 2019) to construct the auxiliary loss. Let $g_\theta(x)$ be the output of the last convolutional neural network (CNN) layer of model $\theta$, immediately before average pooling. Let $g_\theta(x)_i$ be its feature vector at position $i$. For each local update, client $k$ compares global prototype $P_{r-1} = \{P_{r-1}^c | c = 1, 2, ..., N_c\}$ with $g_\theta(x)_i$ and calculates auxiliary loss $L_{\text{aux}}^{S_k}(\theta_r, P_{r-1})$ similar to (2):

$$L_{\text{aux}}^{S_k}(\theta, P_r) = \frac{1}{\sum_{c \in C_k} |S_k(c)|} \sum_{c \in C_k} \sum_{x \in S_k(c)} \sum_i \{d(g_\theta(x)_i, P_r^c)) + \log \sum_{c' \neq c} \exp(-d(g_\theta(x)_i, P_r^{c'})))\}. \tag{3}$$

We also wish to minimize this auxiliary loss, to make the local clients not overly biased and to learn general embedding space.

**Local model update.** Based on the loss functions (2) and (3), for $r \geq 1$, the objective function for local optimization becomes

$$F^{S_k}(\theta_r) = L_{\text{local}}^{S_k}(\theta_r, P_r(k)) + \gamma L_{\text{aux}}^{S_k}(\theta_r, P_{r-1}) \tag{4}$$

where $\gamma$ is a balancing factor. For $r = 0$, we have $F^{S_k}(\theta_r) = L_{\text{local}}^{S_k}(\theta_r, P_r(k))$ since the global prototype is not defined in the first global round. Now based on the the objective function $F^{S_k}(\theta_r)$, each client $k$ performs local update according to Line 16 of Algorithm 1, where $\alpha$ is the learning rate. In federated learning, the clients generally performs multiple local updates, say $E$ times. Hence, the process of local prototype computation (1), loss computations (2), (3) and local model update of Line 16 of Algorithm 1 is repeated $E$ times to obtain $\theta_{r+1}(k)$.

**Model and prototype aggregation at the server.** After performing local updates at the clients, each client $k$ sends its updated local model $\theta_{r+1}(k)$ and the computed local prototypes $P_r(k)$ to the server. Then the server aggregates the local models and the local prototypes according to Lines 18 and 20 in Algorithm 1, respectively, where the weighting factor $\lambda_k = \frac{|S_k|}{\sum_{k=1}^K |S_k|}$ reflects the relative support set sizes.

The above local update and global aggregation processes are repeated for $R$ global rounds ($r = 0, 1, ..., R-1$), and the server finally obtains $\theta_R$ and $P_{R-1}$.

### 3.2.2 ONE-ROUND LOCAL META-UPDATE AND AGGREGATION

Towards the end of each episode processing stage, the clients download $\theta_R$ and $P_{R-1}$ from the server. At each client $k$, the local query loss $L_{\text{local}}^{Q_k}(\theta_R, P_{R-1})$ is calculated similar to (2) based on $Q_k$, $\theta_R$, and $P_{R-1}$. Taking a derivative of $L_{\text{local}}^{Q_k}$ with respect to $\phi_t$ in Line 24 can easily be done through the chain rule, yielding (dropping super/subscripts and terms not critical for now):

$$\nabla_{\phi_t} L(\theta_R) = \nabla_{\theta_R} L(\theta_R) \times \frac{\partial}{\partial \theta_0} \theta_R = \nabla_{\theta_R} L(\theta_R) \times \left( \prod_{r=0}^{R-1} \sum_{k=1}^{K} \lambda_k \frac{\partial}{\partial \theta_r} (\theta_r - \alpha \nabla_{\theta_r} F^{S_k}(\theta_r)) \right),$$

where $\theta_0 = \phi_t$ and the loss function $F^{S_k}$ is the local loss seen in Line 16. We notice that the product term here involves double derivatives, which can be safely ignored according to our empirical observation (which is consistent with what was observed in a parallel setting in MAML of (Finn et al., 2017)). Thus, we simply use first-order approximation $\nabla_{\phi_t} L(\theta_R) \approx \nabla_{\theta_R} L(\theta_R)$ to perform the local meta-update in Line 24. The server finally aggregates the meta-updated models from all clients and moves on to the next episode.

### 3.3 DEPLOYMENT (TESTING)

In the actual deployment or test phase, given a set of clients with unseen classes, the server sets $\theta_0 = \phi_T$ and then performs $R$ rounds of federated learning to obtain $\theta_R$ and $P_{R-1}$. Now, given a test sample, we make prediction based on $\theta_R$ and $P_{R-1}$: we first compute the output of model $\theta_R$ with the new test sample, and then compare the distances with all global prototypes in $P_{R-1}$ to make the decision.

## 4 EXPERIMENTAL RESULTS

We validate our proposed algorithm on two benchmark datasets for meta-learning, *mini*ImageNet (Vinyals et al., 2016) and *tiered*ImageNet (Ren et al., 2018), which have significantly larger numbers of classes than MNIST or CIFAR-10 commonly used in federated learning studies. Following the data splits introduced in (Ravi & Larochelle, 2017), 100 classes are divided into 64 training, 20 test and 16 validation classes for *mini*ImageNet. For *tiered*ImageNet, the dataset is divided into 351/160/97 classes for train/test/validation, respectively. Experimental results on CIFAR-100 dataset can be found in the Supplementary Materials.

**Comparison schemes.** We compare our algorithm with the following schemes. First, as a simplest baseline, we consider FedAvg (McMahan et al., 2017), where a randomly initialized model is trained for $R$ communication rounds. Preparing stage is not utilized for this scheme. Second, we consider a FedAvg-based fine-tuning method, where the overall model is first pre-trained on the training set in the preparing stage, and then fine-tuned to the unseen test set for $R$ communication rounds via FedAvg in the deployment phase. For example, in *mini*ImageNet, a 64-way classifier model is pre-trained first. Next, the last linear layer is replaced and newly initialized, and then the overall model is fine-tuned during testing. Finally, we consider fine-tuning based on one-shot FL (Guha et al., 2019), where the local models are sampled and aggregated by the ensemble cross-validation (CV) method. We allow a larger number of available clients for this scheme to accommodate user sampling. The model is first pre-trained on the training set in the preparing stage, and then fine-tuned to the unseen classes with the scheme of (Guha et al., 2019) for $R$ rounds. For our few-round learning (FRL) algorithm, we utilize both linear classifier and distance-based classifier (Snell et al., 2017) for comparison. For the linear classifier case, we connect an additional linear layer behind CNN layers, as in other baselines. The distance-based classifier utilizes the output of CNN layers, e.g., to build prototypes, instead of using the linear layer. For the distance-based classifier, we observe the effect of global prototype-assisted learning (GPAL) using the auxiliary loss.

**Experimental setup.** We meta-train as well as test our model in federated learning setups. We typically set the target number of communication rounds to $R = 3$. Hence, in the meta-training phase, each episode of our scheme requires 4 global rounds: 3 rounds of local updates and aggregation, and 1 round of local meta-update and aggregation. For a fair comparison, we let all baselines to consume the same amount of communication resources in the meta-training or pre-training phase: $3.2 \times 10^4$ communication rounds between the server and clients. Hence, our scheme is meta-trained over $8.0 \times 10^3$ episodes, taking 4 rounds in each episode. To construct each episode, the server selects $K = 10$ clients in the system for participation. We first consider a 5-way setup where 5 classes are randomly sampled from the train/test sets to construct the datasets of clients in each episode. This type of modeling reflects the practical federated learning setup where a large number of clients having different local data coexist in the system. 120 images are sampled from each class

Table 1: Test accuracies after $R = 3$ communication rounds in a 5-way setup.

| Methods | *mini*ImageNet | | *tiered*ImageNet | |
|---|---|---|---|---|
| | IID | Non-IID | IID | Non-IID |
| FedAvg | 35.44% | 29.79% | 40.98% | 34.26% |
| Fine-tuning via FedAvg | 57.93% | 38.21% | 60.69% | 50.52% |
| Fine-tuning via one-shot FL (Guha et al., 2019) | 60.59% | 36.75% | 62.44% | 50.26% |
| **FRL**: Linear classifier (Ours) | 68.33% | 58.18% | 69.71% | 61.56% |
| **FRL**: Distance-based classifier (Ours) | 72.37% | 66.42% | 74.51% | 68.12% |
| **FRL**: Distance-based classifier + GPAL (Ours) | **73.71**% | **67.33**% | **75.32** % | **68.81**% |

(a) IID scenario, *mini*ImageNet          (b) Non-IID scenario, *mini*ImageNet

Figure 2: Test accuracy after $R = 3$ global rounds in the testing phase, with varying numbers of communication rounds in the meta-training phase (or pre-training phase).

and distributed to $K = 10$ clients depending on two data distribution setups: IID and non-IID. In the IID setup, the data samples from each class are equally distributed to $K = 10$ clients; each client has 12 samples for 5 classes, a total of 60 samples. In the non-IID setup, following the procedure of (McMahan et al., 2017), we first sort the 600 training samples of each episode by its category. We then divide the sorted data into 20 shards of size 30 and randomly allocate 2 shards to each client. Hence, no client would be given data corresponding to more than 2 distinct classes. For both IID and non-IID setups, each client uses one half of its data from each class as support samples, and the remaining half as query samples. In other words, in an IID setup, each client has 6 support samples and 6 query samples for each class. In the non-IID setup, the clients with 2 classes have 15 support samples and 15 query samples for each class, while the clients with 1 class have 30 support samples and 30 query samples for that class. For the one-shot FL method described above, we allow 20 clients and the server samples $K = 10$ of them to aggregate. In the actual deployment (testing) phase, the support set is utilized for $R = 3$ rounds of local updates and the server calculates the test accuracy with the global model/prototype and the gathered query sets of all clients. We averaged the test accuracies over $5.0 \times 10^3$ randomly sampled episodes in the testing phase.

**Implementation details.** For a fair comparison, our method and other baselines share the same neural network architecture. The model follows the settings of (Vinyals et al., 2016), which contains 4 consecutive $3 \times 3$ convolutional layers with 32 filters. Successively, each CNN output goes through ReLU activation, batch normalization and $2 \times 2$ max pooling. We adopt the Adam optimizer as the meta-learner with a learning rate of $\beta = 0.01$ and an SGD optimizer as the learner with a learning rate of $\alpha = 0.1$. The meta-learning rate reduces by one-tenth after $2.0 \times 10^4$ rounds. We set the number of local epochs to $E = 1$. As mentioned, we do not see advantages in using the second derivatives in computing the gradient in Line 24 of Algorithm 1 in the meta-updating process. Given the memory and computational overhead issues as well, we naturally disregard the second derivatives and apply first-order approximation.

**Experimental results in a 5-way setup.** Table 1 shows the test accuracies in a 5-way setup. The model is meta-trained targeting $R = 3$ and tested after $R = 3$ rounds in the test phase. First, for both IID and non-IID scenarios, it can be seen that FedAvg yields significantly lower accuracy compared to others, since it uses a randomly initialized model for training. By pre-training the model for $3.2 \times 10^4$ communication rounds, we observe that FedAvg-based fine-tuning gives significant performance gains compared to naive application of FedAvg. The fine-tuning scheme based on one-shot federated learning shows further performance improvements in the IID setup. However, since $K = 10$ clients are sampled from 20 clients for this ensemble method, there possibly exist some unseen classes when building the global model/prototype in the non-IID setup, which lowers the performance compared to fine-tuned FedAvg. Our few-round learning algorithm performs the best, with the distance-based classifier showing better accuracy compared to the linear classifier. It can be also seen that the performance of the global model can be further improved by global

Table 2: Test accuracies after $R = 3$ communication rounds in a random-way (3~7) setup.

| Methods | *mini*ImageNet | | *tiered*ImageNet | |
|---|---|---|---|---|
| | IID | Non-IID | IID | Non-IID |
| FedAvg | 36.98% | 31.32% | 41.35% | 35.34% |
| Fine-tuning via FedAvg | 62.27% | 53.04% | 63.62% | 51.11% |
| Fine-tuning via one-shot FL (Guha et al., 2019) | 63.52% | 51.92% | 64.36% | 49.49% |
| **FRL**: Linear classifier (Ours) | 65.06% | 53.51% | 63.48% | 54.87% |
| **FRL**: Distance-based classifier (Ours) | 73.06% | 66.97% | 72.36% | 69.07% |
| **FRL**: Distance-based classifier + GPAL (Ours) | **74.32**% | **67.94**% | **73.41**% | **69.86**% |

Table 3: Test accuracies after $R = 1$ communication round in a 5-way setup.

| Methods | *mini*ImageNet | | *tiered*ImageNet | |
|---|---|---|---|---|
| | IID | Non-IID | IID | Non-IID |
| FedAvg | 31.64% | 28.86% | 36.14% | 32.56% |
| Fine-tuning via FedAvg | 50.58% | 36.76% | 51.42% | 41.16% |
| Fine-tuning via one-shot FL (Guha et al., 2019) | 52.07% | 32.05% | 52.40% | 40.74% |
| **FRL**: Linear classifier (Ours) | 62.61% | 51.09% | 64.52% | 55.78% |
| **FRL**: Distance-based classifier (Ours) | 72.10% | 66.13% | 74.37% | 67.95% |
| **FRL**: Distance-based classifier + GPAL (Ours) | **72.58**% | **66.63**% | **74.69**% | **68.15**% |

prototype-assisted learning with the auxiliary loss. Fig. 2 shows how the final test accuracy (after 3 fixed rounds) in the deployment phase increases as the number of communication rounds in the meta-training (or pre-training) phase grows. The overall results in Table 1 and Fig. 2 confirm the advantage of exploiting meta-learning and global prototype-assisted learning ideas to facilitate few-round federated learning.

**Experimental results in a random-way setup.** In Table 2, we provide test accuracies in a more general random-way setup. To construct each episode in the meta-training phase, $\tau$ classes are randomly and independently sampled from the train sets, where $\tau$ follows a discrete uniform distribution in the range [3, 7]. The same procedure of random $\tau$ sampling is also applied during the test phase. As in the 5-way setup, we sample 120 images from each class to construct an episode, having a total of $120\tau$ samples. In the IID setup, each client has 12 samples for $\tau$ classes. In the non-IID setup, we again divide the sorted data into 20 shards of size $6\tau$ and randomly allocate 2 shards to each client. To meta-train the linear classifiers in this random-way setup, we fix the output size of last linear layer to 7 and exploit perceptron from the beginning, according to the selected $\tau$ value. The results are consistent with the one in the 5-way setup, confirming significant advantages of our few-round learning algorithm even in a more general setup.

**Experimental results with mismatch on $R$.** In Table 3, we show the results for mismatched $R$. The model is meta-trained targeting $R = 3$ as in Tables 1 and 2, but the actual test accuracies are evaluated after $R = 1$ communication round in the test phase. Since the global model is obtained after only one global round, the accuracies are lower than in Table 1. Overall, it can be seen that our few-round learning algorithm is still powerful in this mismatch scenario. Testing on mismatch with $R = 2$, as provided in Supplementary Materials, also shows consistent results.

**Comparison with personalization scheme.** We note that our formulation targets creating a *global* model while the previous works on federated meta-learning (Lin et al., 2020; Chen et al., 2018) aim at personalized *local* models. Given these different goals, in a non-IID setup, our method can generally handle broader classes of data than existing personalization approaches. However, there is a simple way to obtain a global model based on these personalization models, in case a need arises for a globalized model after-the-fact: just run rounds of local updates and aggregations starting from the local models but using data now collected across interested clients. We were curious about how this globalized model would fare. In Supplementary Materials, we added experimental results for this scheme. A quick conclusion is that the performance is comparable to the best fine-tuning methods but lags well behind our methods targeting a global model from the get-go.

## 5 CONCLUSION

We proposed a meta-learning strategy that enables few-round learning. Given a set of clients with new tasks, our meta-trained model generalizes well, within only a few communication rounds be-

tween the server and the clients. Extensive experimental results confirm the significant advantages of our method over different baselines in both IID and non-IID data distribution setups.

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

## A  ADDITIONAL EXPERIMENTS UNDER MISMATCH ON $R$

In Table A.1, we show additional experimental results on a mismatch scenario in the 5-way setup, where the model is meta-trained targeting $R = 3$ but tested after only $R = 2$ communication rounds in the deployment phase. Our few-round learning algorithm consistently achieves higher accuracies compared to other baselines in this mismatch scenario.

Table A.1: Meta-trained targeting $R = 3$. Accuracy evaluated after $R = 2$ rounds in a 5-way setup.

| Methods | *mini*ImageNet | | *tiered*ImageNet | |
|---|---|---|---|---|
| | IID | Non-IID | IID | Non-IID |
| FedAvg | 33.93% | 30.35% | 39.14% | 34.49% |
| Fine-tuning via FedAvg | 55.37% | 34.43% | 57.80% | 48.05% |
| Fine-tuning via one-shot FL | 57.71% | 34.35% | 59.76% | 46.97% |
| **FRL**: Linear classifier (Ours) | 67.08% | 54.70% | 68.59% | 60.38% |
| **FRL**: Distance-based classifier (Ours) | 72.33% | 66.28% | 74.34% | 68.04% |
| **FRL**: Distance-based classifier + GPAL (Ours) | **73.51**% | **66.86**% | **74.97**% | **68.63**% |

## B  ADDITIONAL EXPERIMENTS WITH TARGET $R = 2$

While the results shown in the main manuscript utilized the model targeting $R = 3$, here we show results with target $R = 2$, in the same 5-way setup. The model is meta-trained with $R = 2$. The accuracies are evaluated after $R = 2$ communication rounds in Table B.1, and evaluated after $R = 1$ communication round in Table B.2. Overall results show that our few-round learning algorithm outperforms other baselines in this $R = 2$ case as well.

Note that Table B.1 below and Table A.1 above evaluate accuracies after $R = 2$ global rounds, while the models are meta-trained targeting $R = 2$ and $R = 3$, respectively. Comparing these results, we can say that meta-training using a higher $R$ does not provide performance advantage in the deployment (testing) phase. Hence, given a constraint on $R$ in the deployment time, there is no need to perform "higher-round training", which saves the communication resources utilized in the meta-training phase.

Table B.1: Meta-trained targeting $R = 2$. Accuracy evaluated after $R = 2$ rounds in a 5-way setup.

| Methods | *mini*ImageNet | | *tiered*ImageNet | |
|---|---|---|---|---|
| | IID | Non-IID | IID | Non-IID |
| FedAvg | 33.93% | 30.35% | 39.14% | 34.49% |
| Fine-tuning via FedAvg | 57.83% | 37.15% | 58.61% | 42.61% |
| Fine-tuning via one-shot FL | 58.44% | 37.05% | 58.97% | 40.81% |
| **FRL**: Linear classifier (Ours) | 63.45% | 57.11% | 66.39% | 61.29% |
| **FRL**: Distance-based classifier (Ours) | 72.31% | 66.39% | 74.44% | 68.02% |
| **FRL**: Distance-based classifier + GPAL (Ours) | **73.22**% | **67.11**% | **74.82**% | **68.53**% |

Table B.2: Meta-trained targeting $R = 2$. Accuracy evaluated after $R = 1$ round in a 5-way setup.

| Methods | *mini*ImageNet | | *tiered*ImageNet | |
|---|---|---|---|---|
| | IID | Non-IID | IID | Non-IID |
| FedAvg | 31.64% | 28.86% | 36.14% | 32.56% |
| Fine-tuning via FedAvg | 48.60% | 36.44% | 47.58% | 35.89% |
| Fine-tuning via one-shot FL | 48.69% | 36.52% | 48.16% | 35.71% |
| **FRL**: Linear classifier (Ours) | 60.71% | 55.75% | 63.99% | 59.61% |
| **FRL**: Distance-based classifier (Ours) | 72.25% | 66.38% | 74.33% | 68.31% |
| **FRL**: Distance-based classifier + GPAL (Ours) | **72.63**% | **67.09**% | **74.56**% | **68.93**% |

Table C.1: Comparison with personalization scheme in a 5-way setup. Meta-trained targeting $R = 3$, and tested after $R = 3$ rounds.

| | *mini*ImageNet | | *tiered*ImageNet | |
| --- | --- | --- | --- | --- |
| **Methods** | IID | Non-IID | IID | Non-IID |
| Personalization: Linear Classifier | 54.05% | 54.05% | 60.87% | 50.21% |
| Personalization: Distance-based Classifier | 61.69% | 53.66% | 64.78% | 55.98% |
| **FRL**: Linear classifier (Ours) | 68.33% | 58.18% | 69.71% | 61.56% |
| **FRL**: Distance-based classifier (Ours) | 72.37% | 66.42% | 74.51% | 68.12% |
| **FRL**: Distance-based classifier + GPAL (Ours) | **73.71**% | **67.33**% | **75.32** % | **68.81**% |

Table C.2: Comparison with personalization scheme in a random-way (3∼7) setup. Meta-trained targeting $R = 3$, and tested after $R = 3$ rounds.

| | *mini*ImageNet | | *tiered*ImageNet | |
| --- | --- | --- | --- | --- |
| **Methods** | IID | Non-IID | IID | Non-IID |
| Personalization: Linear Classifier | 53.51% | 46.16% | 59.89% | 49.83% |
| Personalization: Distance-based Classifier | 62.44% | 53.93% | 62.28% | 53.45% |
| **FRL**: Linear classifier (Ours) | 65.06% | 53.51% | 63.48% | 54.87% |
| **FRL**: Distance-based classifier (Ours) | 73.06% | 66.97% | 72.36% | 69.07% |
| **FRL**: Distance-based classifier + GPAL (Ours) | **74.32**% | **67.94**% | **73.41**% | **69.86**% |

## C    COMPARISON WITH THE PERSONALIZATION SCHEME

In this section, we provide additional experimental results to compare our few-round learning algorithm with the personalization schemes (Lin et al., 2020; Chen et al., 2018). For the personalization scheme, the clients perform local meta-updates with their query sets right after the local updates with the support sets, in the meta-training phase. For testing, local updates and aggregations are repeated for $R$ global rounds to construct a global model after-the-fact. Tables C.1 and C.2 compare our scheme with the personalization scheme combined with global aggregation, in a 5-way setup and a random-way (3∼7) setup, respectively. The detailed settings are exactly the same as in the main manuscript. The personalization scheme combined with global aggregation achieves comparable performance with best fine-tuning methods in the main manuscript, but falls well below our schemes geared to few-round FL.

## D    ADDITIONAL EXPERIMENTS WITH LESS DATA SAMPLES AT EACH CLIENT

So far, we sampled 120 images from each class to construct an episode. In this section, we observe the effect of reducing the number of data samples in each client. Instead of sampling 120 images, we now sample 60 images from each class to construct an episode. In a 5-way setup, 60·5/10 = 30 samples are allocated to each client. In the IID setup, the data samples from each class are equally distributed to $K = 10$ clients; each client has 6 samples for each of 5 classes, having a total of 30 samples. For each class, 3 samples are utilized as support samples and other 3 samples are utilized as query samples. In the non-IID setup, we again divide the sorted data into 20 shards of size 15 and randomly allocate 2 shards to each client.

Table D.1 shows the test accuracies in this setup. Again we consider a 5-way setup, and the model is meta-trained targeting $R = 3$. The accuracies are evaluated after $R = 3$ rounds of federated learning. Since we utilize less data samples for training, the performances are degraded compared to the results in the main manuscript. But, overall, the trend is consistent with the results we observed so far, confirming the advantage of our few-round learning algorithm utilizing meta-learning.

## E    ADDITIONAL EXPERIMENTS ON CIFAR-100

To confirm the applicability of our proposed scheme further, we provide additional experiments on CIFAR-100 in Table.E.1. Similar to *mini*ImageNet, we split 100 classes in CIFAR-100 into 64

Table D.1: Performance with less data samples at each client (3 shots per class per client for IID as compared to our usual setup of 6 shots in this work) in a 5-way setup. Meta-trained targeting $R = 3$, and tested after $R = 3$ rounds.

| Methods | *mini*ImageNet | | *tiered*ImageNet | |
|---|---|---|---|---|
| | IID | Non-IID | IID | Non-IID |
| FedAvg | 34.39% | 29.08% | 39.61% | 33.48% |
| Fine-tuning via FedAvg | 57.70% | 38.09% | 60.08% | 48.61% |
| Fine-tuning via one-shot FL | 60.36% | 38.18% | 61.63% | 47.79% |
| **FRL**: Linear classifier (Ours) | 65.46% | 55.78% | 67.32% | 59.61% |
| **FRL**: Distance-based classifier (Ours) | 70.04% | 63.08% | 71.96% | 65.25% |
| **FRL**: Distance-based classifier + GPAL (Ours) | **70.81**% | **63.48**% | **72.54**% | **65.67**% |

Table E.1: Test accuracies after $R = 3$ communication rounds in 5-way setup on CIFAR-100.

| Methods | CIFAR-100 | |
|---|---|---|
| | IID | Non-IID |
| FedAvg | 45.13% | 37.50% |
| Fine-tuning via FedAvg | 65.92% | 41.02% |
| Fine-tuning via one-shot FL | 66.22% | 36.53% |
| **FRL**: Linear classifier (Ours) | 70.96% | 58.09% |
| **FRL**: Distance-based classifier (Ours) | 80.14% | 72.54% |
| **FRL**: Distance-based classifier + GPAL (Ours) | **80.35**% | **72.66**% |

training, 16 validation and 20 test classes. The results are consistent with the tables on other datasets, confirming the advantage of our approach.

# F ADDITIONAL EXPERIMENTS WITH A LARGER NUMBER OF CLIENTS

To demonstrate the scalability of our proposed method, we performed additional experiments with $K = 50$ clients. Table F.1 shows the details, which indicates that our scheme outperforms other baselines in a larger scale federated learning system.

Table F.1: Performance with $K = 50$ clients. Accuracies are obtained after $R = 3$ communication rounds in 5-way setup.

| Methods | *mini*ImageNet | |
|---|---|---|
| | IID | Non-IID |
| FedAvg | 36.85% | 30.75% |
| Fine-tuning via FedAvg | 63.20% | 36.09% |
| **FRL**: Linear classifier (Ours) | 70.01% | 63.88% |
| **FRL**: Distance-based classifier (Ours) | 73.89% | 73.02% |
| **FRL**: Distance-based classifier + GPAL (Ours) | **74.03**% | **73.22**% |

