# OpenReview forum: "Few-Round Learning for Federated Learning"
_ICLR.cc/2021/Conference — Reject_

### Official Review · AnonReviewer2 · 2020-10-28
**This is a federated prototypical network.**

**Rating:** 3
**Confidence:** 5

**Review:**

The paper is to train a meta-model in a small number of selected nodes in a federated learning environment, and then use the meta-model to assist the federated learning in reducing the communication rounds. It is basically a federated version of a prototypical network.

The proposed method relies on a strong assumption that there is a meta-training environment in federated learning. It is not a standard FL setting. Moreover, given the assistance of the meta-model, there is no guarantee that the federated learning environment will converge in a few-round.

The major technique contribution of the proposed method is how to meta-train a global model in a federated setting. In particular, it adapts the prototypical network to fit the federated setting. It is unclear how the proposed method provides any theoretical contribution rather than applied research.

In the experiment, one dataset is not enough to support the effectiveness of the proposed method.
More federated learning-related benchmark datasets should be discussed, e.g., FeMNIST, Shakespeare texts, CIFAR, and FeCelebA.

In particular, the proposed two-stage procedure is equivalent to: learn a global model in a standard FL setting, and then conduct personalized deployment for each device or a specific group of devices. Therefore, in the experiment part, the authors need to add more baseline methods, for example, some personalized federated learning method should be selected as baseline methods.

THE MAJOR CONCERN: In Algorithm 1, lines 16 and 18 are a federated aggregation-based updating, and line 24 is a prototypical-based meta learner updating. These two updating methods are inconsistent which are to optimize different objectives, and the authors should give an overall loss to unify the updating steps rather than force two kinds of updating into one framework.

Typo:
“Metra-training” in Figure 1.

---

> ### Author Response · Authors · 2020-11-18
> **Response to Reviewer 2**
>
> $\textbf{Practical issues and convergence guarantee:}$ Our experimental setup reflects the practical federated learning scenario where a large number of clients with different local data coexist in the system. The setup to handle unseen tasks after only a few FL rounds is also realistic. In each episode of meta-training, the server selects a set of clients to participate. Here, to model the local datasets of clients in each episode, we sampled several classes from a miniImageNet or tieredImageNet, which have a significantly larger number of classes compared to the datasets that considered in the previous works on federated learning. This is actually a more challenging environment compared to previous works considering MNIST, CIFAR-10 with only 10 classes, and this type of modeling reflects the practical federated learning scenarios where a large number of clients having different local data coexist in the system.  Moreover, as we mentioned in our original manuscript, if the server has some collected data, there is no need to perform federated learning with the distributed clients for meta-training; the server can construct various episodes and simply mimic the federated learning setup by itself, to obtain a meta-trained initial model. So there are more than one ways (by federated learning or by mimicking the process at the server) to obtain a meta-trained model in real-world applications. Regarding the theoretical guarantee, please note that analysis has been tricky even for the widely-known schemes in few-shot learning, such as MAML and Prototypical Networks. We provided extensive experimental results to validate our statements as done in the previous works on meta-learning.
>
> $\textbf{Theoretical contribution:}$ We first note that our meta-update process follows the spirit of MAML. In addition, motivated by Prototypical Networks, we utilized distance-based classifier for the models, and prototype aggregation methods at each round. As you might agree, on the subject of meta-learning, theoretical analysis is tricky and even for the widely-used schemes such as MAML and Prototypical Networks, theoretical rigor has been absent.
> We would like to also note that our results are not simple applied research. We emphasize that we are the first team of researchers to establish that once we successfully adopt a meta-learning strategy in the preparation stage to obtain an appropriate initial model, few-round learning is possible in the deployment phase for an arbitrary set of clients wishing to predict unseen data. In addition to starting a new paradigm of FL, we believe there are significant algorithmic innovations that are interesting and obviously new as well. Note that previous attempts exist to link meta learning with federated learning, but to our knowledge all these efforts consider more “predictable” applications of meta learning to provide initialization for local optimizations at individual devices.
>
> $\textbf{Benchmark datasets for federated learning:}$ We utilized miniImageNet and tieredImageNet for experiments which have a large number of classes. This is actually a more challenging dataset compared to the benchmarks on federated learning. We performed additional experiments with CIFAR-100 (which also has a relatively large number of classes compared to others) and obtained consistent results (please refer to the supplementary materials of the revised manuscript).
>
> $\textbf{Comparison with the personalized federated learning scheme:}$ As can be seen in page 8 of our original manuscript, we already compared our scheme with the personalized federated learning method. A more detailed results can be found in the supplementary material, which was also included in our submitted manuscript. To sum up, our approach is superior to existing individualized device optimization followed by meta-learning. We note that the meta-training environment of these personalized federated learning schemes are the same as ours, which is practical.
>
> $\textbf{Regarding the two-step procedure for meta-training:}$ Meta-training generally has a two-step procedure. First is the adaptation step with the support sets and second is the meta-update step with the query sets. Here, we note that these two steps are performed in serial and should be separated to describe the meta-training procedure clearer. The first step corresponds to lines 16 and 18 in our algorithm, where R rounds of federated learning is performed with the support sets to mimic the deployment phase. Here, each local update is performed to minimize the local support loss of each client. After repeating lines 16 and 18 for R global rounds, the server obtains $\theta_R$. Now based on this $\theta_R$, the goal of the meta-update process is to solve the following:  $\phi^*= \underset{\phi}{argmin}\sum_{k=1}^K \rho_kL^{Q_k}(\theta_R)$. This is the reason why we have a serial process of lines 16, 18 and line 24.

---

### Official Review · AnonReviewer1 · 2020-10-28
**Review #1**

**Rating:** 5
**Confidence:** 4

**Review:**

## Summary

This paper proposes a new paradigm to train federated learning models. In particular, following the spirit of meta-learning for few-shot learning, the authors propose to meta-train an initial model so that starting from this point, only $R$ (eg, 3) rounds of FL are needed to produce a satisfying test accuracy.

## Pros
1. The authors made significant efforts in designing the meta-learning strategy for few-round FL.
2. The proposed algorithm has the potential to redefine FL training paradigm. But there should be more validations. My questions and concerns are stated in the next section.

## Cons
The major concern I have is about the way they construct the dataset and evaluate the algorithm. The training task the authors selected is more like a meta-learning standard setting and is not common in federated learning. So I doubt its performance in realistic FL settings. It would be great if the authors can evaluate their algorithm in a standard FL dataset, otherwise it is not convincing.
1. When constructing the meta-learning datasets for each episode, the authors sample several classes from the whole dataset  and then simulate 10 clients based on the selected samples. However, in FL setting, this is infeasible, as the server cannot access the whole dataset. The authors should describe how to construct the meta-learning procedure given hundreds or even thousands of clients without accessing their local data. For example, Shakespeare dataset has 715 train clients and 715 test clients. How to construct the meta-learning procedure from this decentralized data and how the algorithm performs are unclear.
2. The scale of FL is relatively small. At each episode, there is only 10 clients. However, in practical on-device FL, there can be thousands of clients for training and testing. For example, in [1], StackOverflow has 342,477 training clients and 204,088 test clients. Even EMNIST dataset has 3400 test clients. The performance of the proposed algorithm is unclear in these realistic large-scale FL problems.
3. The meta-train algorithm require the computation of full-batch loss at each round, which consumes more computational resources than vanilla FedAvg. The authors are supposed to discuss this additional overhead.

## Post-rebuttal comments
Thanks the authors for the response! I've read it and other reviewers' comments. I feel the authors didn't directly answer my questions and just reiterate what they have in the paper. Unfortunately, it is still unclear to me how to perform meta-training on standard FL training tasks, for example, shakespeare in [1]. In this training task, there're total 700+ clients. Does that mean in the meta-training phase, we need to sample 700+ clients for each episode? How to construct this meta-train dataset from a standard federated dataset?

[1] Reddi et al. Adaptive Federated Optimization. 2020

---

> ### Author Response · Authors · 2020-11-18
> **Response to Reviewer 1**
>
> $\textbf{Constructing episodes in practice:}$ First of all, we note that the server does not have access to any datasets in our setup. The server just selects a set of clients in the system to construct each episode. Here, we sample several classes (5 classes for 5-way setup and a random number of classes for random-way setup) to just model the classes in the specific episode (classes in the client in a specific episode). This type of modeling is one way of reflecting the practical federated learning scenarios where a large number of clients having different local data coexist in the system.
> Further and importantly, if the server has some collected data, there is no need to perform federated learning with the distributed clients for meta-training; the server can construct various episodes and simply mimic the federated learning setup by itself, to obtain a meta-trained initial model. We mentioned this in our original manuscript.
> In short, our meta-trained model can be obtained in different ways (by federated learning or by mimicking the process at the server) in real-world applications.
>
> $\textbf{Scalability:}$ We added experiments with a larger number of clients (K=50) and obtained consistent results. Please refer to the supplementary materials of the revised manuscript.
>
> $\textbf{Regarding the full-batch loss:}$ We note that mini-batch update is also possible for our scheme, by computing the local prototype based on the samples in each mini-batch. For example, in a 5-way setup, one can use mini-batch size of 15 having 3 samples for each class instead of the full batch case having 6 samples for each class.

---

### Official Review · AnonReviewer4 · 2020-10-28
**The paper needs more clarification to explain the proposed complex setting and procedure.**

**Rating:** 4
**Confidence:** 4

**Review:**


This paper studied the combination of federated learning tasks in a meta-learning setting. In particular, with the assistance of the pre-trained meta-model, the new FL model's training can be completed within limited communication rounds. It was inspired by the meta-learning method used in few-shot learning scenario. This paper proposed a few-round learning (FRL) algorithm and designed global prototype-assisted learning (GPAL) scheme to assist training. It is an interesting topic to combine meta-learning with federated learning.

The weaknesses of this paper are summarized below.

1. The proposed method updates meta-model in each client. However, the meta-learning task consumes lots of computation resources and highly relies on the large number of classes. These make it hard to train a meta-model in a local client in a federated system. Although the setting sounds useful, it is hard to realize in real-world applications.

2. This paper is relevant to two widely-known few-shot learning methods, MAML, and prototypical network. So, it is better to consider MAML+FL and/or ProtoNet+FL as baselines to make the proposed methods more convincing and prove the efficacy of the proposed loss functions.

3. Given the complexity of the proposed algorithm and associated hyperparameters, the authors could anonymously release the source code in the reviewing stage. More details about the experimental platform used in this paper should be given.

4. As illustrated in the Experimental setup on page 6, the meta/pre-training phase needs a large number of communication rounds. Is it appropriate for the bandwidth-limited or time-sensitive applications? Will this be a distracter in few-round learning scenarios?

5. For the 5-way setup in Table 1, there are 5 classes are randomly sampled from the dataset in each episode, which means that all the clients contain all the training classes, 64 classes for miniImageNet and 351 classes for tieredImageNet, locally. This is impractical because most local clients only have limited information to share.

6. The representation of trainable parameters in Algorithm 1 is a little bit confusing. For example, \theta and \phi are actually the same parameters. The only difference between them is that \theta is updated during local update using the support set, while \phi is updated during local meta-update using query set. Since the algorithm is an important part of this paper, the definition and use of these parameters should be much clearer. If possible, the authors can add a detailed interpretation of these two parameters.

---

> ### Author Response · Authors · 2020-11-18
> **Response to Reviewer 4 (Part 1)**
>
> $\textbf{Applicability of our scheme in practice:}$ Let us first clarify our problem setup and goal. We focus on preparing an initial model that would allow inference after just a few quick rounds of FL. The assumed environment is one where local clients must handle tasks that were not seen during the model preparation stage. We note that our experimental setup reflects the practical federated learning scenario where a large number of clients having different local data coexist in the system. In each episode, the server selects a set of clients to participate. Since the selected clients aid meta-training for R communication rounds only in a specific episode, the burden of individual clients would be significantly smaller compared to conventional FL. Moreover, the computational burden at each client can be further reduced by first-order approximation of subsection 3.2.2. Here, we note it is not necessary to have a large number of classes in each client. It is also not necessary to have a large number of classes in each episode (i.e., in the set of clients in each episode). Moreover, as we mentioned in our original manuscript, if the server has some collected data, there is no need to perform federated learning with the distributed clients for meta-training; the server can construct various episodes and simply mimic the federated learning setup by itself, to obtain a meta-trained initial model. Given an initial model after the meta-training phase, in the deployment phase, the service provider can provide this model to the clients to enable few-round learning.
>
> We do note that handling unseen tasks is a tough challenge in general, let alone in the already tricky FL environment. Nevertheless, we feel this is an important step towards the right direction – namely, to address the need for distributed clients to handle unseen tasks using only a quick rounds of FL.
>
> $\textbf{Comparison with meta-learning based FL schemes:}$ We would like to first note that our meta-update process follows the spirit of MAML. Moreover, motivated by Prototypical Networks, we utilized distance-based classifier for the models, and prototype aggregation methods at each round. In the experiment section, the scheme “FRL: Linear classifier (Ours)” can be viewed as MAML+FL. It is hard to apply PN+FL targeting R-round federated learning since the conventional PN does not update the model (only computes the prototypes) with the support, which means that the model should not change for R global rounds. Hence, we modified the scheme to update the distance-based classifier with the support set, which corresponds to “FRL: Distance-based classifier (Ours)”.
>
> $\textbf{Source code:}$ The source code is made available in supplementary material.
>
> $\textbf{Applicability in bandwidth-limited or time-sensitive applications:}$ The meta-training phase (or the preparing stage) and the deployment phase should be distinguished. The meta-training phase is a step that service provider prepares the initial model, and there are various ways to obtain this meta-trained model (by federated learning or by mimicking the process of federated learning at the server) as discussed above. Now given a meta-trained initial model, in the deployment phase, the service provider can provide this model to the clients for fast federated learning, especially to the clients who are targeting bandwidth-limited or time-sensitive applications. The reviewer touches upon a very important point, though. While gearing for a few-round leaning is highly useful for bandwidth-limited applications, we note that pre-training (meta-training) can possibly take place over low-bandwidth routes or during low-traffic time zones.
>
> $\textbf{Episode construction in practice:}$ First of all, we note that reviewer’s argument here is not quite true; the clients do not have to contain all training classes in its locality. In our setting, the server selects a set of clients to construct each episode in the meta-training phase. Here, we sample a number of classes to just model the classes in the specific episode (classes in the client in a specific episode). The 5-way setup is the case where there are 5 classes in the datasets of the clients participating in a specific episode; each client can have 1 to 5 classes in its local dataset. To consider a more general and a practical setup, we also considered a random-way setup where the number of classes in the client set in a specific episode is random. These types of modeling are a reasonable way to reflect the practical federated learning scenarios where a large number of clients having different local data coexist in the system. Moreover, as we mentioned above, the server can construct each episode with its collected data and simply mimic the federated learning setup/procedure. Hence, there are different options in constructing each episode in practical setups.

---

> > ### Author Response · Authors · 2020-11-18
> > **Response to Reviewer 4 (Part 2)**
> >
> >
> > $\textbf{Difference between $\phi$ and $\theta$:}$ We utilized different notations $\theta$ and $\phi$ to emphasize the meta-update process. Here, $\phi$ is a parameter that is updated by meta-update process with the query sets, and $\theta$ is a parameter that is updated during R rounds of federated learning with the support sets. We utilized these two parameters to describe our algorithm clearer. As the reviewer suggests, we added clear descriptions about these two parameters.

---

### Official Review · AnonReviewer3 · 2020-10-30
**This work attempts to reduce the communication cost in federated learning by applying meta-learning. The proposed method requires a meta-training phase and a few-round training phase. The authors demonstrate its effectiveness using a dataset in the context of meta-learning to simulate the dataset in federated learning.**

**Rating:** 4
**Confidence:** 5

**Review:**

1. Strength:

Targeting an important problem of FL: reducing the communication cost.


2. Weakness:

This work simply applies the meta-learning method into the federated learning setting. I can’t see any technical contribution, either in the meta-learning perspective or the federated perspective. The experimental results are not convincing because the data partition is not for federated learning. Reusing data partition in a meta-learning context is unrealistic for a federated learning setting.

The title is misleading or over-claimed. Only the adaptation phase costs a few rounds, but the communication cost of the meta-training phase is still high.

The non-IID partition is unrealistic. The authors simply reuse the dataset partitions used in the meta-learning context, which is not a real federated setting. Or in other words, the proposed method can only work in the distribution which is similar to the meta-learning setting.

Some meta earning-related benefits are intertwined with reducing communication costs. For example, the author claimed the proposed method has better generalization ability, however, this is from the contribution of the meta-learning. More importantly, this property can only be obvious when the data distribution cross-clients meet the assumption in the context of meta-learning.

The comparison is unfair to FedAvg. At least, we should let FedAvg use the same clients and dataset resources as those used in Meta-Training and Few-Rounds adaptation.

“Episodic training” is a term from meta-learning. I suggest the authors introduce meta-learning and its advantage first in the Introduction.

Few-shot FL-related works are not fully covered. Several recent published knowledge distillation-based few-shot FL should be discussed.



3. Overall Rating

I tend to clearly reject this paper because: 1) the proposed framework is a simple combination of meta-learning and federated learning. I cannot see any technical contribution. 2) Claiming the few round adaptations can reduce communication costs for federated learning is misleading, since the meta-training phase is also expensive. 3) the data partition is directly borrowed from meta-learning, which is unrealistic in federated learning.

---------after rebuttal--------

The rebuttal does not convince me with evidence, thus I keep my overall rating. I hope the author can obviously compare the total cost of meta-learning phase plus FL fine-tuning phase with other baselines.

---

> ### Author Response · Authors · 2020-11-18
> **Response to Reviewer 3 (Part 1)**
>
> $\textbf{Technical contribution in meta-learning or federated learning perspective:}$ Our work is certainly about putting meta-learning and federated learning together, but in a way that has never been anticipated before, despite the practical significance of our new problem formulation – namely, to prepare an initial model that allows inference on unseen tasks after only a few rounds of FL. At a more specific level, our algorithm provides episodic training based on successful adaptation of MAML-style optimization to FL mixed with Prototype-assisted learning. In addition to starting a new paradigm of FL, we believe there are significant algorithmic innovations that are interesting and obviously new as well. Note that previous attempts exist to link meta-learning with federated learning, but to our knowledge all these efforts consider more “predictable” applications of meta learning to provide initialization for local optimizations at individual devices.
>
> $\textbf{Data partition for meta-learning and federated learning:}$ We remark that our experimental setup reflects the practical federated learning scenario where a large number of clients having different local data coexist in the system. In each episode, the server selects a set of clients to participate. Here, to model the local datasets of clients in each episode, we sample a number of classes (5 classes for 5-way and a random number of classes for random-way) from a miniImageNet or tieredImageNet, which have a significantly larger number of classes compared to the datasets that considered in the previous works on federated learning. This is actually a more challenging environment compared to previous works considering MNIST, CIFAR-10 with only 10 classes, and this type of modeling is a good way to reflect the practical federated learning scenarios where a large number of clients having different local data coexist in the system. To confirm the applicability of our scheme further, we performed additional experiments with the CIFAR-100 dataset and obtained consistent results (please refer to supplementary materials of our revised manuscript). Importantly, as we mentioned in our original manuscript, if the server has some collected data, there is no need to perform federated learning with the distributed clients for meta-training; the server can construct various episodes and simply mimic the federated learning setup by itself, to obtain a meta-trained initial model.
>
> $\textbf{Regarding the over-claimed terminology on few-round learning:}$ Note that our goal is to prepare an initial model that can quickly obtain a reliable global model within R rounds of federated learning. Here, the key issue is that the tasks of the clients participating in FL were not known in the preparation stage. How should we prepare the model that generalizes well for an arbitrary set of clients with unknown tasks? Following the spirit of meta-learning for few-shot learning, we tried to solve this practically meaningful but challenging problem via meta-learning. Please note that few-shot learning also requires a large number of “shots” in the meta-training phase. We are a drawing an interesting analogy between “few-shot learning,” and our need to pre-train a model to tee up for FL over just a few rounds. We stress that for fair comparison, we utilized the same amount of communication resources for pre-training the baseline schemes (please refer to the descriptions on schemes “Fine-tuning via FedAvg” and “Fine-tuning via one-shot FL” in our experiment section). Moreover, as we mentioned above, the server can simply mimic the federated learning setup and procedure with its collected data. In this case, meta-training phase does not require any communication between the server and the clients. We tried to make all these points clearer in the revised manuscript.
>
> $\textbf{Dataset partition for meta-learning setup:}$ We feel that the non-IID partition is not only realistic but actually a necessary assumption to reflect the reality of federated learning with a large number of participants with varying sets of local data. Please refer to our response to your second comment above (our explanation of how we prepared our episodic training data and why it makes sense). We are not interested in categorizing meta-learning problems and federated learning problems. We are just trying our best to reflect the reality of federated learning in preparing our data sets.
>
> $\textbf{Generalization ability:}$ Yes, the generalization ability comes from meta-learning. This is exactly what we attempted (and succeeded). We are the first researchers to show that once a meta-learning strategy is adopted successfully in the preparation stage to obtain an appropriate initial model, few-round learning is possible in the deployment phase for an arbitrary set of clients. Regarding the data preparation for meta-learning, please refer to our responses to your second comment above.

---

> > ### Author Response · Authors · 2020-11-18
> > **Response to Reviewer 3 (Part 2)**
> >
> > $\textbf{Fair comparison with FedAvg:}$ We note that “Fine-tuning via FedAvg” is exactly the same scheme as one that the reviewer is referring to. The same clients and communication resources are utilized to pre-train the model. This is why it has better performance compared to the naïve application of FedAvg in Tables 1,2,3. Moreover, it can be observed from the tables that our scheme provides significant advantages compared to the baseline that the reviewer is suggesting. For example, it can be seen that the proposed method surpasses “Fine-tuning via FedAvg” by 14.63 % in an IID setup with tieredImageNet.
> >
> > $\textbf{Describing meta-learning in the Introduction:}$ Please note that we already explained the advantage of meta-learning in Section 1 (Introduction). A more detailed description on episodic training is in Section 2 (Related Works).
> >
> > $\textbf{Related works on few-shot FL:}$ We strived to cite all we know, including the unpublished archived manuscripts. The authors of (Guha et al., 2019) (the paper that we already covered in related work section) utilize knowledge distillation for semi-supervised learning setup and utilize ensemble methods for supervised learning setup. Since we target supervised learning, we simply described this paper using the terminology of ensemble, not knowledge distillation. We would appreciate if the review could point us to the specific papers we are missing.
> >
> > (Guha et al., 2019) Neel Guha, Ameet Talwalkar, and Virginia Smith. One-shot federated learning. arXiv preprint arXiv:1902.11175, 2019..
> >
> > $\textbf{Response to the comments on overall rating:}$ We respectfully disagree in that 1) Our frame work that brings together meta-learning and federated learning is neither simple nor anticipated, as evidenced by the lack of relevant prior attempts despite the practical significance. 2) The goal here is to deal with unseen tasks (this was perhaps unclear in the original manuscript) and thus the meta-training phase is inevitable; however, when we make a fair comparison taking into account all resources including those taken in meta-training our method gives superior results compared to other methods. 3) Our data participation reflects a realistic FL scenario where a large number of participants with distinct datasets coexist in the distributed network.

---

### Author Response · Authors · 2020-11-18
**General comments to reviewers**

We thank you for your efforts and constructive suggestions. We have worked them into the revised version of the paper. Many of your comments have been particularly helpful in clarifying our problem formulation and objectives.

First, here we should clearly state our problem setup and goal. While we deal with federated learning (FL), we are not after standard FL problems. We focus on preparing an initial model that would allow inference after just a few quick rounds of FL. The assumed environment is one where local clients must handle tasks that were not seen during the model preparation stage. In the context of image data, the clients will be able to predict unseen classes after only a few rounds of FL. This type of setting is practical as can be seen in the case of medical image processing: once a model is pre-trained to be able to diagnose a number of known types of diseases, this model would then be used to classify images corresponding to unseen types of diseases only after a few FL rounds of further training.

Our main question is, how should we prepare the model that generalizes well for an arbitrary set of clients with unknown tasks? To tackle this problem, we take a meta-learning strategy to prepare the initial model so that once this meta-training phase is over, only R rounds of FL would produce a model that will satisfy the needs of all participating clients. There are two ways to prepare this initial model in practice.

First is to perform federated learning with a new set of clients selected for each episode. To model the local datasets of clients in each episode during pre-training (or meta-training in our context), we sample a number of classes (5 classes for 5-way and a random number of classes for random-way) from the dataset. This type of modeling is a good way to reflect the practical federated learning scenarios where a large number of clients having different local data coexist in the system. Secondly, if the server has some collected data, there is no need to perform federated learning with distributed clients for meta-training; the server can construct various episodes and simply mimic the federated learning setup by itself, to obtain a meta-trained initial model.

Once the meta-training phase is over, the service provider offers the obtained initial model to the clients that participated in the pre-training stage (who used one set of data to participate in meta-training but wants to predict another set of data) or to a new set of clients with new tasks. An arbitrary set of clients having this initial model can now obtain a reliable global model within only a few rounds of federated learning.

We provide our answers below in more detail to each question raised.

---

### Decision · Program_Chairs · 2021-01-07
**Final Decision**

**Decision:**

Reject

**Comment:**

This paper proposes a meta-learning based few-shot federated learning approach to reduce the communication overhead incurred in aggregating model updates. The use of meta-learning also gives some generalization benefits. The reviewers think that the paper has the following main issues (see reviews for more details):
* Limited technical novelty - the paper seems to simply combine meta-learning with federated learning
* Not clear whether the communication overhead is actually reduced because the meta-learning phase can require significant communication and computation.
* The experimental evaluation, in particular, the data distribution, could have been more realistic.

I hope that the authors can use the reviewers' feedback to improve the paper and resubmit to a future venue.